# Hardiness and Expectations for Future Life: The Roles of Perceived Stress, Music Listening for Negative Emotion Regulation, and Life Satisfaction

**DOI:** 10.3390/bs13100852

**Published:** 2023-10-18

**Authors:** Alexander Park, Kyung-Hyun Suh

**Affiliations:** 1Smith College of Liberal Arts, Sahmyook University, Seoul 01795, Republic of Korea; hjpark@syu.ac.kr; 2Department of Counseling Psychology, Sahmyook University, Seoul 01795, Republic of Korea

**Keywords:** hardiness, stress, music listening, life satisfaction, expectations for future life

## Abstract

This study investigated the relationship between hardiness and Korean adults’ expectations for future life, and verified the multiple mediating effects of perceived stress, music listening for negative emotion regulation, and life satisfaction on that relationship. The participants were 412 Korean adults aged 20–65 years. PROCESS Macro 3.5 Model 80 was used to examine the multiple mediating effects. Correlational analysis showed that hardiness was positively correlated with music listening for negative emotion regulation, life satisfaction, and expectations for future life, whereas it was negatively correlated with perceived stress. Perceived stress was negatively correlated with life satisfaction and expectations for future life, whereas music listening for negative emotion regulation was positively correlated with life satisfaction and expectations for future life. In the multiple mediation model, the relationships between hardiness and expectations for future life, the sequential mediating effect of perceived stress and life satisfaction, and the sequential mediating effect of music listening for negative emotion regulation and life satisfaction were significant. The direct effect of hardiness on expectations for future life was also significant, indicating that perceived stress, music listening for negative emotion regulation, and life satisfaction only partially mediated the relationship between hardiness and expectations for future life. It seems, thus, that perceived stress, music listening for negative emotion regulation, and life satisfaction play an important role in Korean adults’ expectations for future life.

## 1. Introduction

Aristotle defined happiness as “the ultimate goal of human existence” in Nicomachean Ethics [1]. This means that people live to be happy. Diener introduced subjective well-being as a psychological variable representing happiness, defining it as being satisfied with one’s life and experiencing more pleasant and fewer unpleasant emotions [2]. Meanwhile, the concept of life satisfaction, which is cognitive rather than affective, is the overall satisfaction with one’s life and is known to be closely related to happiness [3]. However, in some cases, one’s expectations for future life, rather than life satisfaction, may be more closely associated with happiness [4,5]. An optimistic view of own well-being in the face of an uncertain future can improve quality of life [6]. Accordingly, in this study, we attempted to verify a model that could predict people’s expectations for their future life.

Suh found that expectations for future life, rather than life satisfaction, correlated with happiness, especially for college students and young people who were living difficult lives and sacrificing their present lives while preparing for a satisfactory future [7]. Meanwhile, Pavot et al. found that while young people were less satisfied with their current lives than older people, they had higher expectations for a satisfactory future life [8]. Because expectations for future life may have a positive effect on people’s lives as much as life satisfaction, Kim et al. argued that expectations for a satisfied life in the future should be included when measuring subjective well-being [5]. Still, there is also evidence that if individuals are satisfied with their present lives, they may be more likely to have higher expectations for their future lives. In previous studies, life satisfaction was positively correlated with expectations for future life [7,9]. Therefore, this study hypothesized that satisfaction with present life can affect expectations for future life.

In this study, we chose hardiness as a variable that could affect life satisfaction and expectations for future life. Psychological hardiness is a personal trait that buffers the harmful effects of stressors, maintains quality of life, and improves well-being [9,10,11,12]. Hardiness, a concept introduced by Kobasa, is a personality trait that can help individuals respond resiliently to stressors and recover from harmful reactions to stressors [13]. Hardiness was conceptualized as the tendency to involve oneself in activities of one’s own life; to believe that one can influence the events and circumstances around oneself; to prefer change rather than stability and to be motivated for opportunities for personal growth rather than threats to security [13]. Hardiness is a dispositional personality trait that develops early in life and is relatively stable over time and dependent on circumstances [14]. Therefore, we assumed that hardiness could stably predict life satisfaction and expectations for future life.

Stress is the most well-known threat to subjective well-being [9,15]. As aforementioned, hardiness can theoretically make individuals less affected by stressors, thereby maintaining or improving their well-being [13], and protect individuals from cardiovascular diseases, which are most commonly caused by stressors [16,17]. In a previous study, hardiness was found to be positively correlated with life satisfaction, whereas it was negatively correlated with perceived stress [18]. Perceived stress is defined as the feelings or thoughts people have about how much stress they are experiencing during a certain period of time [19]. In Suh et al.’s study, the hardiness of Korean people was found to be closely related to life satisfaction, sharing 34.8% (*r* = 0.59) of the variance with life satisfaction [20]. Accordingly, this study assumed that perceived stress may mediate the relationships of hardiness with life satisfaction or expectations for future life.

In addition to perceived stress, this study assumed that music listening for negative emotion regulation might mediate the relationship between hardiness and life satisfaction or expectations for future life. In previous studies, music engagement, such as listening to music and attending musical events, was positively correlated with subjective well-being, including life satisfaction [21,22], and Schunk et al. [23] found that emotion regulation was positively correlated with life satisfaction in college students. Wang and Suh also demonstrated that the emotional adaptability derived through music listening, including the regulation of negative emotions, was positively correlated with life satisfaction among older adults [24]. For this reason, Reybrouck and Eerola described music listening as a eudaimonic enjoyment beyond hedonic pleasure [25]. Vuoskoski and Eerola found that even the feeling of sadness may lead individuals to enjoy sad music and experience pleasure through the intensifying feelings of being moved [26]. In this study, music listening for negative emotion regulation refers to the adaptive function of music listening that relieves anger and anxiety.

Because hardy people are good at controlling themselves and actively using coping strategies, they may prefer music listening as a coping strategy to control negative emotions [13,27,28]. Coping refers to active and intentional strategies used to reduce negative emotions in difficult situations [29]. Reybrouck et al. emphasized that music listening as a coping mechanism is a complex process that requires all one’s sensory, physiological, behavioral, and cognitive aspects, involving understand of one’s environment; we assumed that the aforementioned characteristics of hardiness would lead people to use music listening to regulate negative emotions [30]. Therefore, hardy people may prefer music listening for negative emotion regulation, and as a result, they may be satisfied with their present lives and have higher expectations for future life.

In this study, it was hypothesized that hardiness directly decreases the perceived stress of Korean adults, whereas it directly increases their music listening for negative emotion regulation, life satisfaction, and expectations for future life. That is, we hypothesized that perceived stress and music listening for negative emotion would mediate the relationships of hardiness with life satisfaction or expectations for future life in Korean adults. This hypothesized multiple mediation model, thus, includes the sequential double mediating effect of perceived stress and life satisfaction on the relationship of hardiness and expectations for future life; it also encompasses the sequential double mediating effect of music listening for negative emotion regulation and life satisfaction on the relationship between hardiness and expectations for future life in Korean adults (see Figure 1).

## 2. Methods

### 2.1. Participants

In total, 412 Korean adults aged 20–65 years participated in this study. The data were collected by Embrain, an online research company, in March 2022. Of the participants, 201 (49.2%) were men and 217 (50.8%) were women. The average age of participants was 44.60 (standard deviation [SD], 13.46) years, and a total of 81 participants were in their 20s (19.0%), 84 were in their 30s (19.7%), 86 were in their 40s (20.1%), 86 were in their 50s (20.1%), and 90 were over 60 years old (21.1%). 

### 2.2. Measures

#### 2.2.1. Hardiness

Hardiness was measured using the Brief Measure of Hardiness developed for Korean people by Suh [31]. This 12-item scale was developed to provide a reliable and valid measure of hardiness, a personal trait that enables efficient recovery from stress. It comprises three four-item factors, as follows: commitment, self-directedness, and tenacity. Items are responded to using a six-point Likert scale ranging from 1 (not at all true) to 6 (very true), with higher scores indicating stronger hardiness. In this study, only individual total scores were included in the analysis, and the internal consistency (Cronbach’s α) of commitment, self-directedness, tenacity, and total score were 0.88, 0.85, 0.87, and 91, respectively.

#### 2.2.2. Perceived Stress

The perceived stress experienced by the participants was measured using the Perceived Stress Scale developed by Cohen and Williamson [19]. This study used the Korean version of the scale validated by Lee et al. [32], comprising 10 items, five of which should be reversed scored. Each item is rated on a five-point Likert scale ranging from 1 (never) to 5 (very often), with higher scores indicating greater perceived stress. In this study, the Cronbach’s α of the scale was 0.82.

#### 2.2.3. Music Listening for Negative Emotion Regulation

To measure music listening for negative emotion regulation, we used subscales of the adaptive functions of music listening scale developed by Groarke and Hogan [33]. This scale consists of 46 items that measure the stress regulation, strong emotional experiences, rumination, sleep, reminiscence, anger regulation, anxiety regulation, awe and admiration, loneliness regulation, cognitive regulation, and the identity regulation functions of music listening. For this study, we used only the subscales of anger regulation and anxiety regulation, each with seven items. Items are rated on a five-point Likert scale ranging from 1 (strongly disagree) to 5 (strongly agree). The Cronbach’s α of anger regulation, anxiety regulation, and the total scale were 0.90, 0.92, and 95, respectively, in this study.

#### 2.2.4. Life Satisfaction

Participants’ life satisfaction was measured using the Satisfaction with Life Scale developed by Diener et al. [34]. This scale was developed to measure one’s global cognitive judgments of own life. Life satisfaction differs from subjective well-being in that its judgment is driven more by cognition than emotion. Life satisfaction can be assessed for a specific aspect of life or for life in general. The five items are rated from 1 (strongly disagree) to 7 (strongly agree), and the Cronbach’s α of this scale was 0.88 in this study.

#### 2.2.5. Expectations for Future Life

To measure expectations for future life, we used the Life Satisfaction Expectancy Scale developed by Kim [4]. It consists of five modified items from the Satisfaction with Life Scale [34]. For example, “The conditions of my life are excellent” was modified to “In the future, the conditions of my life will be better”. The five items are rated from 1 (strongly disagree) to 7 (strongly agree), and the Cronbach’s α of the scale in this study was 0.96.

### 2.3. Procedure

Prior to data collection, the study was approved by our Institutional Review Board (protocol code: SYU 2022-02-007), and all procedures were conducted according to ethical guidelines. Before responding to the online survey, the participants provided written consent. Participants were also informed that even after agreeing to participate in the survey, they could withdraw their participation at any time if they experienced any type of psychological discomfort. In addition, we informed the participants that all data would only be used for research purposes and would be stored on an encrypted computer for 3 years and then discarded.

### 2.4. Statistical Analysis

Data were analyzed using IBM SPSS Statistics^®^ version 25 and PROCESS Macro 3.5. The mean, standard deviation, skewness, and kurtosis of the data were calculated using SPSS Statistics, and Pearson’s product–moment correlation analysis was performed. We used PROCESS Macro 3.5 Model 80 to verify the multiple mediating effects [35]. Additionally, to examine the significance of the mediation model, we used bootstrapping with a 95% confidence interval and 5000 replications. Bootstrapping, a non-parametric resampling method, can examine the mediating effect even if the sample distribution does not meet normality [35]. The multiple mediation model analyzed in this study was adjusted according to participant age.

## 3. Results

### 3.1. Relationship between the Variables Involved in the Expectation for Future Life

Table 1 shows the results of the correlational analysis for participants’ hardiness, perceived stress, music listening for negative emotion regulation, life satisfaction, and expectations for future life. The absolute values of the skewness and kurtosis of the variables included in the analysis did not exceed 2.0, and the variance of all variables was close to a normal distribution, which satisfied the conditions for parametric statistical analysis [36].

Correlational analysis revealed that the participants’ age was negatively correlated with perceived stress (*r* = −0.111, *p* < 0.05) and expectations for future life (*r* = −0.168, *p* < 0.001). Hardiness was positively correlated with music listening for negative emotion regulation (*r* = 0.285, *p* < 0.001), life satisfaction (*r* = 0.518, *p* < 0.001), and expectations for future life (*r* = 0.594, *p* < 0.001), and negatively correlated with perceived stress (*r* = −0.327, *p* < 0.001). Perceived stress was also negatively correlated with life satisfaction (*r* = −0.449, *p* < 0.001) and expectations for future life (*r* = −0.333, *p* < 0.001). 

However, music listening for negative emotion regulation was positively correlated with life satisfaction (*r* = 0.195, *p* < 0.001) and expectations for future life (*r* = 0.162, *p* < 0.001). Furthermore, life satisfaction was positively correlated with expectations for future life (*r* = 0.669, *p* < 0.001), and shared approximately 44.8% of the variation.

### 3.2. Verification of the Multiple Mediation Model for Expectations for Future Life

This study analyzed the multiple mediating effects of perceived stress, music listening for negative emotion regulation, and life satisfaction on the relationship between hardiness and expectations for future life (see Table 2). Prior to the analyses, we checked for any potential multicollinearity in the data. If the tolerance is less than 0.2 and the variance inflation factor is greater than 5.0, multicollinearity problems may occur [37]. In this study, the tolerance of the predictors ranged from 0.638–0.898, the variance inflation factors were 1.113–1.568; thus, there was seemingly no significant multicollinearity issue. In addition, the Durbin–Watson value was 1.951, which is close to 2.0, indicating that autocorrelation was not detected in this sample.

The results showed that hardiness positively influenced music listening for negative emotion regulation (*B* = 0.341, *p* < 0.001) and life satisfaction (*B* = 0.266, *p* < 0.001), and negatively influenced perceived stress (*B* = −0.206, *p* < 0.001) in this model. Music listening for negative emotion regulation positively influenced life satisfaction (*B* = 0.057, *p* < 0.05), but did not significantly directly influence expectations for future life (*B* = −0.010, n.s.). Moreover, perceived stress negatively influenced life satisfaction (*B* = −0.380, *p* < 0.001), but did not significantly directly influence expectations for future life (*B* = −0.024, n.s.). Additionally, life satisfaction positively influenced expectations for future life (*B* = 0.527, *p* < 0.001). 

Figure 2 shows that participants’ perceived stress (*β* = −0.020, n.s.) and music listening for negative emotion regulation (*β* = −0.016, n.s.) cannot significantly influence their expectations for future life without an indirect effect on life satisfaction. This means that perceived stress can eventually lower life satisfaction and lead to the loss of hope for the future, and music listening for negative emotion regulation leads to life satisfaction and expectations for future life. The direct effect of hardiness on expectations for future life was reduced in this model, but remained statistically significant. These results indicate that perceived stress, music listening for negative emotion regulation, and life satisfaction partially mediate the relationship between hardiness and expectations for future life.

Using 5000 bootstrap replications, we examined the multiple mediating effects of perceived stress, music listening for negative emotion regulation, and life satisfaction in the relationship between hardiness and expectations for future life (Table 3). The total indirect effect was 0.194 (0.1427–0.2463), which was significant because there was no zero difference between the upper and lower bounds at the 95% confidence interval.

However, the simple mediating effect of the path from hardiness to expectations for future life via music listening for negative emotion regulation (−0.0207~0.0131) and via perceived stress (−0.0162~0.0257) were not significant. In contrast, the path via life satisfaction was significant (0.1002–0.1848). 

Additionally, the sequential double mediating effect of perceived stress and life satisfaction on the relationship between hardiness and expectations for future life (H→PS→LS→EFL) was 0.041 (0.0240–0.0632) and significant. The sequential double mediating effect of music listening for negative emotion regulation and life satisfaction on the relationship between hardiness and expectations for future life (H→ML→LS→EFL) was 0.010 (0.0005–0.0236) and significant. 

We also examined whether there were any differences in the indirect effect sizes revealed in this study (Table 3). The effect sizes of the indirect paths of H→LS→EFL were greater than that of the indirect paths of H→PS→LS→EFL and H→ML→LS→EFL. In addition, the double mediating effect size of perceived stress and life satisfaction on the relationship between hardiness and expectations for future life (H→PS→LS→EFL) was significantly greater than that of music listening for negative emotion regulation and life satisfaction (H→ML→LS→EFL: −0.0535~−0.0105).

## 4. Discussion

This study identified the relationships between hardiness, perceived stress, music listening for negative emotion regulation, life satisfaction, and expectations for future life in Korean adults. Furthermore, it verified the multiple mediating effects of perceived stress, music listening for negative emotion regulation, and life satisfaction on the relationship between hardiness and expectations for future life. The findings of this study provide valuable information and knowledge for researchers and mental health professionals invested in supporting people’s quality of life. The implications of the findings of this study are discussed below. 

In this study, the age of participants was not significantly correlated with life satisfaction but was negatively correlated with perceived stress and expectations for future life. This suggests that as Korean adults age, their satisfaction with their present lives does not change much, but they perceive less stress and have fewer expectations for their future. Prior scholars had already shown that older adults tend to perceive daily hassles as less stressful than younger people [38,39]. This result indicates that as age increases, people perceive less stress, their level of life satisfaction does not change much, and they become more likely to have fewer expectations about their future. Shrira et al. found that as age increases, expectations of living standards improve slightly, but the expectation of deterioration increases significantly [40]. Considering that the aforementioned deterioration may result in a decrease in overall expectations for future life, it is necessary to analyze this topic in greater depth in future studies.

People with higher hardiness scores were more satisfied with their present lives and expected more from their future lives in this study. Specifically, hardiness was closely correlated with life satisfaction and expectations for future life, and shared approximately 35.3% (*r* = 0.594) of the variation in expectations for future life. Depressed patients have very low expectations for the future [41], and this study provides a rationale for the clinical application of the hardiness training proposed by Maddi et al. to depressed people [27]. Particularly, the results of the current study depicted that the stronger the hardiness, the lower the perceived level of stress, and the shared variance between these two variables was approximately 10.7% (*r* = 0.327). The fact that hardiness accounted for more than 10% of the variance in perceived stress indicates that hardiness training may be clinically applicable for stress management.

Moreover, in this study, perceived stress and life satisfaction sequentially mediated the relationship between hardiness and expectations for future life. As hypothesized, hardiness can make Korean adults perceive less stress, which makes them more satisfied with their lives, and potentially have expectations for their future. In this mediation model, the direct effect of perceived stress on expectations about future life was not significant. This means that Korean adults’ perceived stress only influences expectations for future life through making them dissatisfied with their present lives. This mediation model reveals that life satisfaction plays a determinant role and perceived stress plays an important role—albeit to a lesser extent—in the relationship between hardiness and expectations for future life.

In the proposed multiple mediation model, we hypothesized a sequential double mediating effect of music listening for negative emotion regulation and life satisfaction on the relationship between hardiness and expectations for future life. In this model, people with higher hardiness scores were more likely to listen to music to regulate their negative emotions. Hardiness is correlated with active rather than passive coping [42], and, indeed, hardy persons in our sample used music as a coping resource to regulate negative emotions. Moreover, our model shows that music listening for negative emotion regulation only influences expectations for future life through life satisfaction. 

Additionally, the indirect effect of life satisfaction on the relationship between hardiness and expectations for future life was greater than any double mediating effect. This means that it is necessary to pay attention to the path through which hardiness affects life satisfaction without experiencing perceived stress and music listening for negative emotion regulation. Moreover, as the direct effect of hardiness on expectations for future life was significant, it is necessary to study how hardiness affects expectations for future life in the absence of its mediation of life satisfaction.

## 5. Conclusions and Limitations

In this study, Korean adults’ expectations for future life were positively correlated with hardiness, music listening for negative emotion regulation, and life satisfaction, and negatively correlated with perceived stress. The results indicate the determinant role of life satisfaction in the relation between hardiness and expectations for future life in this population. Additionally, in a multiple mediation model, the indirect effects of music listening for negative emotion regulation and life satisfaction, as well as the indirect effects of perceived stress and life satisfaction on the relationship between hardiness and expectations for the future life in Korean adults, were significant. 

This study has the following limitations. First, the sample that responded to the online survey is not representative of adults in Korea or worldwide. Therefore, the results of this study must be confirmed using samples from various countries and more representative samples of the Korean population. Second, although we assumed a causal relationship between the variables based on the results of previous studies and logic, our correlational study cannot confirm cause-and-effect relationships. Future experimental and longitudinal studies are warranted to confirm the causality relationships between the study variables. Third, this study did not consider the type of music with regard to music listening. However, since physiological evidence found that sad music can also move individuals and cause them to experience positive emotions [43], most kinds of music may be effective when used to regulate emotions positively. There may be individual differences in such effects [44]; it is necessary to explore individual differences in the emotional adaptive function of music listening in further studies. Finally, all variables in this study were measured using self-reported questionnaires. Responsiveness to self-reported questionnaires on quality of life has been shown to potentially differ according to age [45]. Therefore, it is necessary to study the relationships among the variables of interest in the current study using various measurement methods in the near future. 

## Figures and Tables

**Figure 1 behavsci-13-00852-f001:**
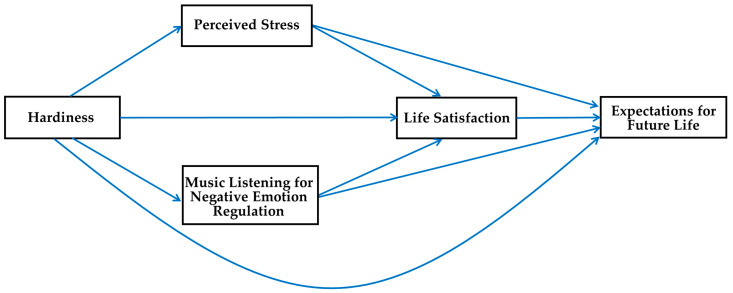
Proposed multiple mediation model.

**Figure 2 behavsci-13-00852-f002:**
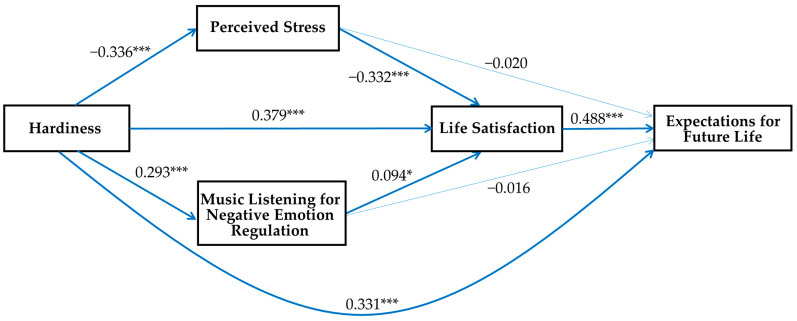
The multiple mediation model of perceived stress, music listening for negative emotion regulation, and life satisfaction on hardiness and expectations for future life (* *p* < 0.05, *** *p* < 0.001; standardized coefficients).

**Table 1 behavsci-13-00852-t001:** Correlational matrix of hardiness, perceived stress, music listening for negative emotion regulation, life satisfaction, and expectations for future life (*N* = 412).

Variables	1	2	3	4	5	6
1. Age	1					
2. Hardiness	−0.068	1				
3. Perceived stress	−0.111 *	−0.327 ***	1			
4. Music listening for regulation of negative emotion	0.091	0.285 ***	0.005	1		
5. Life satisfaction	−0.033	0.518 ***	−0.449 ***	0.195 ***	1	
6. Expectations for future life	−0.168 ***	0.594 ***	−0.333 ***	0.162 ***	0.669 ***	1
*M*	43.79	53.18	28.36	48.08	19.89	24.32
*SD*	13.01	8.40	5.15	9.79	5.90	6.38
Skewness	−0.24	0.01	0.26	−0.65	−0.11	−0.46
Kurtosis	−1.24	0.09	0.15	1.21	−0.14	0.14

* *p* < 0.05, *** *p* < 0.001.

**Table 2 behavsci-13-00852-t002:** Multiple mediating effects of perceived stress, music listening for negative emotion regulation, and life satisfaction on hardiness and expectations for future life.

Variables	*B*	*S.E.*	*t*	LLCI	ULCI
Mediating variable model (Outcome variable: Music listening for negative emotion regulation)					
Constant	26.275	3.435	7.65 ***	19.5227	33.0279
Hardiness	0.341	0.055	6.20 ***	0.2330	0.4491
Age	0.084	0.036	2.36 *	0.0139	0.1535
Mediating variable model (Outcome variable: Perceived stress)					
Constant	41.646	1.776	23.45 ***	38.1558	45.1370
Hardiness	−0.206	0.028	−7.25 ***	−0.2620	−0.1502
Age	−0.053	0.018	−2.90 **	−0.0893	−0.0171
Mediating variable model (Outcome variable: Life satisfaction)					
Constant	−14.862	2.692	5.52 ***	9.5690	20.1545
Hardiness	0.266	0.031	8.54 ***	0.2049	0.3274
Music listening for negative emotion regulation	0.057	0.025	2.24 *	0.0069	0.1061
Perceived stress	−0.380	0.049	−7.79 ***	−0.4762	−0.2844
Age	−0.024	0.183	−1.32	−0.0601	0.0119
Dependent variable model (Outcome variable: Expectations for future life)					
Constant	4.432	2.547	1.74	−0.5742	9.4384
Hardiness	0.252	0.031	8.15 ***	0.1908	0.3122
Music listening for negative emotion regulation	−0.010	0.023	−0.45	−0.0559	0.0351
Perceived stress	−0.024	0.048	−0.51	−0.1182	0.0695
Life satisfaction	0.527	0.045	11.67 ***	0.4389	0.6167
Age	−0.637	0.017	−3.81 ***	−0.0967	−0.0308

* *p* < 0.05, ** *p* < 0.01, *** *p* < 0.001. Note, LLCI: lower level for confidence interval; ULCI: upper level for confidence interval.

**Table 3 behavsci-13-00852-t003:** Indirect effects of the mediation model.

Path	Effect	*S.E*.	BC 95% CI
Total indirect effect	0.194	0.026	0.1427~0.2463
Ind1: H→ML→EFL	−0.035	0.085	−0.0207~0.0131
Ind2: H→PS→EFL	0.005	0.011	−0.0162~0.0257
Ind3: H→LS→EFL	0.141	0.022	0.1002~0.1848
Ind4: H→ML→LS→EFL	0.010	0.006	0.0005~0.0236
Ind5: H→PS→LS→EFL	0.041	0.010	0.0240~0.0632
Ind1–Ind2	−0.009	0.013	−0.0340~0.0172
Ind1–Ind3	−0.144	0.024	−0.1912~−0.0985
Ind1–Ind4	−0.014	0.010	−0.0358~0.0036
Ind1–Ind5	−0.045	0.013	−0.0718~−0.0189
Ind2–Ind3	−0.135	0.026	−0.1872~−0.0867
Ind2–Ind4	−0.005	0.125	−0.0310~−0.0187
Ind2–Ind5	−0.036	0.016	−0.0707~−0.0088
Ind3–Ind4	0.130	0.023	0.0873~0.1756
Ind3–Ind5	0.099	0.023	0.0555~0.1449
Ind4–Ind5	−0.031	0.011	−0.0535~−0.0105

H = hardiness, ML = music listening for negative emotion regulation, PS = perceived stress, LS = life satisfaction, EFL = expectations for future life, Ind = indirect effect.

## Data Availability

The datasets analyzed in this study are available from the corresponding author upon request.

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
