# Peer review of "Hardiness and Expectations for Future Life: The Roles of Perceived Stress, Music Listening for Negative Emotion Regulation, and Life Satisfaction"

_behavsci, 2023, doi:10.3390/bs13100852_

Round 1

Reviewer 1 Report

From the paper title we see "music listening" but inside we do not see any reference or discussion about what type of music! Please try to solve this problem.  Important to discuss knowing these ideas!

Please have a look at the references and how the references were inserted or not. The level of plagiarism relatively high.

See like example this links: https://www.frontiersin.org/articles/10.3389/fpsyg.2021.745888.

https://www.frontiersin.org/articles/10.3389/fpsyg.2021.663134   indicated by the plagiarism software.

Author Response

Thank you very much for your comments to improve the quality of this articles and we learned a lot.

The revised parts were marked in red, and we included the page and line of the revised part.

Response to Reviewer 1 Comments

Point 1. From the paper title we see "music listening" but inside we do not see any reference or discussion about what type of music!

Response 1: Thank you for your good comment. Music listening in this study includes all music and does not consider differences in types of music. However, regarding to your comment, we added it as a limitation of the study as follows. (Line 334-337)

Third, this study did not consider the type of music regarding to music listening. However, since physiological evidence found that sad music can also move individuals and cause them to experience positive emotions [43], most kinds of music may be effective when used to regulate emotions positively.

Point 2. Please have a look at the references and how the references were inserted or not. The level of plagiarism relatively high.

See like example this links: https://www.frontiersin.org/articles/10.3389/fpsyg.2021.745888.

https://www.frontiersin.org/articles/10.3389/fpsyg.2021.663134 indicated by the plagiarism software

Response 2: Thank you for your advice. The first reference you pointed out is a study that analyzed the relationship between emotional regulation and life satisfaction. Although they used general emotional regulation, not emotional regulation of music listening, as a variable, it is an article that supports the rationale of the relationship between variables in this study, so we cited it. The second reference was a study that analyzed the relationship between multicultural children's hardiness, life satisfaction, and expectation for future life, and was cited because it was an article that supported the rationale of the relationship between the variables of this study. In fact, that article was researched and published by my colleagues and I. However, since the topics are different and the manuscript were written independently, I am confident that only proper citation was made and that there was no plagiarism in the manuscript.

Reviewer 2 Report

The paper makes the hypothesis that the trait of hardiness decreases the degree of perceived stress of adult Korean people and increases their degree of music listening aimed at negative emotion regulation, as well as life satisfaction and expectations for future life. By exploiting existing scales for the involved variables, the authors present the correlations among them and a multiple mediation model of their reciprocal relations.

They find that perceived stress is negatively correlated with the positive variables of hardiness, music listening for negative emotions, life satisfaction and future expectations, while these positively correlate with each other.

Perceived stress decreases satisfaction and expectations, while music listening increases them but life satisfaction is a mediator in both cases. For example, Korean adults’ perceived stress only influences expectations for future life through making them dissatisfied with their present lives.

The results encourage the authors to propose to apply hardiness training in clinical settings.

The paper is a neat study, although not very original: all the scales used, except the one about future expectation,  are taken from previous works; moreover the study is not so conclusive since, as the Authors acknowledge, the cause-effect links among the variables are not clear.

As to the phrasing of the paper, some little change might enhance ease of comprehension.

Two points concern the words “expectations” and “perceived stress”.

When the Authors speak of Expectations, it should be clear from the outset that they mean the term as definitely positive. This is clear later in the paper, but not in the first mentions of the word. Strictly speaking, when one has expectations about the future, they might be either positive or negative (and so they are for pessimistic and depressed people).

A similar point might be made concerning “perceived stress”. There are cases in which perceiving stress might be considered a very positive thing: should one not perceive it vis-à-vis an objectively stressing problem, this might be due to freudian removal, which could be very maladaptive. The Authors should be more precise in this respect.

One minor issue for ready comprehension are the letters used in Fig. 3. If the Authors could use more transparent letters, e.g.  H for hardiness, PS for Perceived stress, M for music listening, S for satisfaction, E for expectations, the figure could be more expressive.

Author Response

Thank you very much for your comments to improve the quality of this articles and we learned a lot.

The revised parts were marked in red, and we included the page and line of the revised part.

Response to Reviewer 2 Comments

Point 1. When the Authors speak of Expectations, it should be clear from the outset that they mean the term as definitely positive. This is clear later in the paper, but not in the first mentions of the word. Strictly speaking, when one has expectations about the future, they might be either positive or negative (and so they are for pessimistic and depressed people).

Response 1: Thank you for your nice comment. As you advised, we added a sentence that our expectations for future life are positive as below. (Line 44-47)

Because expectations for future life may have a positive effect on people’s lives as much as life satisfaction, Kim et al. argued that expectations for a satisfied life in the future should be included when measuring subjective well-being [5].

Point 2. A similar point might be made concerning “perceived stress”. There are cases in which perceiving stress might be considered a very positive thing: should one not perceive it vis-à-vis an objectively stressing problem, this might be due to freudian removal, which could be very maladaptive. The Authors should be more precise in this respect.

Response 2: Thank you for your comment. In 'perceived stress', perceived is a terminology that has nothing to do with consciousness or unconsciousness. However, we added a definition for perceived stress because readers may find it confusing. (Line 68-69)

Perceived stress is defined as the feelings or thoughts people have about how much stress they are experiencing during the certain period of time [19].

Point 3. One minor issue for ready comprehension are the letters used in Fig. 3. If the Authors could use more transparent letters, e.g. H for hardiness, PS for Perceived stress, M for music listening, S for satisfaction, E for expectations, the figure could be more expressive.

Response 3: Thank you very much for your valuable. It was not easy to add notes to the abbreviations of each variable in the figure, so I modified Table 3 as you advised. (Line 244-260)

Table 3. Indirect effects of the mediation model

Path

Effect

S.E

BC 95% CI

Total indirect effect

 0.194

0.026

 0.1427 ~ 0.2463

Ind1: H → ML → EFL

−0.035

0.085

−0.0207 ~ 0.0131

Ind2: H → PS → EFL

 0.005

0.011

−0.0162 ~ 0.0257

Ind3: H → LS → EFL

 0.141

0.022

 0.1002 ~ 0.1848

Ind4: H → ML → LS → EFL

 0.010

0.006

 0.0005 ~ 0.0236

Ind5: H → PS → LS → EFL

 0.041

0.010

 0.0240 ~ 0.0632

Ind1 – Ind2

−0.009

0.013

−0.0340 ~ 0.0172

Ind1 – Ind3

−0.144

0.024

 −0.1912 ~ −0.0985

Ind1 – Ind4

−0.014

0.010

−0.0358 ~ 0.0036

Ind1 – Ind5

−0.045

0.013

 −0.0718 ~ −0.0189

Ind2 – Ind3

−0.135

0.026

 −0.1872 ~ −0.0867

Ind2 – Ind4

−0.005

0.125

 −0.0310 ~ −0.0187

Ind2 – Ind5

−0.036

0.016

 −0.0707 ~ −0.0088

Ind3 – Ind4

 0.130

0.023

 0.0873 ~ 0.1756

Ind3 – Ind5

 0.099

0.023

 0.0555 ~ 0.1449

Ind4 – Ind5

−0.031

0.011

 −0.0535 ~ −0.0105

H = Hardiness, ML = Music listening for negative emotion regulation, PS = Perceived stress, LS = Life satisfaction, EFL = Expectations for future life, Ind: Indirect effect

Additionally, the sequential double mediating effect of perceived stress and life satisfaction on the relation of hardiness and expectations for future life (H → PS → LS → EFL) was 0.041 (0.0240–0.0632) and significant. The sequential double mediating effect of music listening for negative emotion regulation and life satisfaction on the relation of hardiness and expectations for future life (H → ML → LS → EFL) was 0.010 (0.0005–0.0236) and significant.

We also examined whether there were any differences in the indirect effect sizes revealed in this study (Table 3). The effect sizes of the indirect paths of H → LS → EFL was greater than that of the indirect path of H → PS → LS → EFL and H → ML → LS → EFL. In addition, the double mediating effect size of perceive stress and life satisfaction on the relation of hardiness and expectations for future life (H → PS → LS → EFL) was significantly greater than that of music listening for negative emotion regulation and life satisfaction (H → ML → LS → EFL: −0.0535 ~ −0.0105).

Reviewer 3 Report

This is a paper of moderate interest for the readership. It deals with the relationship between hardiness and expectations for future life. Though the findings are clear, the main merit of this paper is the introduction of a less known methodology in music research by using multiple mediating analysis. This is an interesting approach which opens up new perspectives for future research. The paper as a whole is well written, thought the methodology could be explained more clearly and in more intuitive terms. The theoretical background of the paper and the reference list in general is also rather limited. I suggest to accept the paper on condition that some remarks and comments are addressed appropriately. I list some of them below, both as general remarks and detailed comments.

General remarks

·      The English language use and writing style are OK. Quite mature style of writing.

·      The contents and findings are nor really impressive. They confirm some intuitive impressions which are less or more common knowledge. It could make the paper stronger if the authors should clearly state what is new.

·      The major strength of the paper is the number of participants (N = 412) and the used methodology. Mediation analysis is not commonly used in music research. The statistics are convincing, but especially the tables and how to interpret them are not always very clear. Much more efforts must be done to explain the tables with more details.

·      The concept of negative emotion regulation seems to be important for this paper, but is not sufficiently explained.

·      Some generalizations should be explained more in detail: e.g. music engagement, emotion regulation. These are very general categories which must be specified to make sense.

·      Table 3 must be explained more clearly. Provide also a motivation why a bootstrapping procedure was needed, given the large number of participants.

·      The list of references is rather limited. See suggestions for additional references below.

·      The theoretical background is also very limited. More could be said about musical enjoyment, well-being, hedonia and eudaimonia, etc.

Detailed comments

·      Page 1, line 31: additional references and text could be added to discuss somewhat more in depth the concept of happiness. In particular Aristotle’s conception of eudaimonia could be mentioned in this regard (see also suggested additional references below).

·      Page 2, line 66: this is not clear. It seems as if hardiness would increase perceived stress, which is obviously not what is meant here. Please reword or rephrase better to exactly explain how this positive relation should be understood.

·      Page 2, line 71: same remark. Explain more in detail what is meant with negative emotion regulation. Does this mean that listening to music that expresses negative emotions makes listener stronger with respect to hardiness? Please explain more clearly as this is a central point of the paper.

·      Age 2, line 73 and 75: please specify somewhat more in detail the concepts of “music engagement“ (which music, which way of listening?) and “emotion regulation”.

·      Page 2, line 80: please elaborate a little more on the concepts of coping and coping strategies. Add additional references if available (see suggestions below).

·      Page 3, line 124: same remark: listening for negative emotion regulation. What is meant with emotion regulation here: to increase or to decrease the experience of negativity? What kind of regulation is meant exactly. This is not yet clear, though it is very important in the paper.

·      Page 4, line 135: why this choice for cognitive judgments only? Please provide at least some motivation for this limitation.

·      Page 6, line 221: why using bootstrapping technique here, given the large number of participants? PIease motivate a little.

·      Page 7, lines 230 ff: the relation between the text of this paragraph and table 3 is not immediately clear.

·      Page 7, table 3: please explain much better how to interpret the table. What does BC stand for? What is meant with Ind 1, Ind2. Not all readers are familiar with bootstrapping. Some more intuitive description of the why and how could help them to interpret the table. Could it make sense also not to use the letters A, B, C and to replace them with the initials of the terms (H = hardness; ML = music listening, PS = perceived stress; LS = life satisfaction, EFL = expectation for future life)? This is only a suggestion. The table should be also more readable by adding the significance levels (p-value or *, **, ***).

Suggested additional references:

Eerola, T., Vuoskoski, J. K., Kautiainen, H., Peltola, H., Putkinen, V., & Schäfer, K. (2021). Being moved by listening to unfamiliar sad music induces rewardrelated hormonal changes in empathic listeners. Annals of the New York Academy of Sciences, Early online. https://doi.org/10.1111/nyas.14660

Eerola, T., Vuoskoski, J. K., Kautiainen, H., Peltola, H., Putkinen, V., & Schäfer, K. (2021). Being moved by listening to unfamiliar sad music induces rewardrelated hormonal changes in empathic listeners. Annals of the New York Academy of Sciences, Early online. https://doi.org/10.1111/nyas.14660

Reybrouck, M. & Eerola, T. (2022). Musical Enjoyment and Reward: From Hedonic Pleasure to Eudaimonic Listening. Behav. Sci. 2022, 12(5), 154; https://doi.org/10.3390/bs12050154

Reybrouck, M., Podlipniak, P. & Welch, D. (2020). Music Listening as Coping Behavior: From Reactive Response to Sense-Making. Behavioral Sciences, 10(7), 119.

Vuoskoski, J., & Eerola, T. (2011). Measuring music-induced emotion. A comparison of emotion models, personality biases, and intensity of experiences. Musicae Scientiae, 15, 159-173.

Vuoskoski, J. & Eerola, T. (2017). The Pleasure Evoked by Sad Music is Mediated by Feelings of Beind Moved.  Frontiers in Psychology, 8(439).

Author Response

Thank you very much for your comments to improve the quality of this articles and we learned a lot.

The revised parts were marked in red, and we included the page and line of the revised part.

Response to Reviewer 3 Comments

Point 1. Page 1, line 31: additional references and text could be added to discuss somewhat more in depth the concept of happiness. In particular Aristotle’s conception of eudaimonia could be mentioned in this regard (see also suggested additional references below).

Response 1: Thank you for your valuable advice. As you advised, we added the following sentence: (Line 82-83)

For this reason, Reybrouck and Eerola illustrated music listening as an eudaimonic enjoyment beyond hedonic pleasure [25].

Reybrouck, M., Eerola, T. Musical enjoyment and reward: From hedonic pleasure to eudaimonic listening. Behav. Sci. 2022, 12, 154. https://doi.org/10.3390/bs12050154

Point 2. Page 2, line 66: this is not clear. It seems as if hardiness would increase perceived stress, which is obviously not what is meant here. Please reword or rephrase better to exactly explain how this positive relation should be understood.

Response 2: Sorry. It was our mistake. Therefore, we corrected it as follows. (Line 66-67)

In a previous study, hardiness was found to be positively correlated with life satisfaction, whereas it was negatively correlated with perceived stress [18].

Point 3. Page 2, line 71: same remark. Explain more in detail what is meant with negative emotion regulation. Does this mean that listening to music that expresses negative emotions makes listener stronger with respect to hardiness? Please explain more clearly as this is a central point of the paper.

Response 3: Thank you very much for your good comment. As you advised, we added the following sentence: (Line 85-87)

In this study, music listening for negative emotion regulation refers to the adaptive function of music listening that relieves anger and anxiety.

Point 4. Page 2, line 73 and 75: please specify somewhat more in detail the concepts of “music engagement “(which music, which way of listening?) and “emotion regulation”.

Response 4: As you advised, we explained the concept of music engagement that was studied in the previous research we cited as follows, as follows. (Line 76-77)

In previous studies, music engagement, such as listening to music and attending musical events, was positively correlated with subjective well-being, including life satisfaction [21,22]

Point 5. Page 2, line 80: please elaborate a little more on the concepts of coping and coping strategies. Add additional references if available (see suggestions below).

Response 5: Thank you for your comment. We added an explanation of coping, including the reference you provided, as follows. (Line 90-95)

Coping refers to active and intentional strategies used to reduce negative emotions in difficult situations [29]. Reybrouck et al. emphasized that music listening as a coping is a complex process that considers all sensory, physiological, behavioral, and cognitive aspects involving understand of one's environment, we assumed that the aforementioned characteristics of hardiness would lead people to use music listening to regulate negative emotions [30].

Billings, A.G., Moos, R. (June 1981). The role of coping responses and social resources in attenuating the stress of life events. J. Behav. Med. 1981, 4, 139–157. https://doi.org/10.1007/BF00844267

Reybrouck, M., Podlipniak, P., Welch, D. Music listening as coping behavior: From reactive response to sense-making. Behav. Sci. 2020, 10, 119. https://doi.org/10.3390/bs10070119

Point 6. Page 3, line 124: same remark: listening for negative emotion regulation. What is meant with emotion regulation here: to increase or to decrease the experience of negativity? What kind of regulation is meant exactly. This is not yet clear, though it is very important in the paper.

Response 6: Thank you for your valuable comment. We included what you advised as below. (Line 112-114)

Vuoskoski and Eerola found that even the feeling of sadness may lead individuals to enjoy sad music and experience pleasure through intensifying feelings of being moved [26].

Third, this study did not consider the type of music regarding to music listening. However, since physiological evidence found that sad music can also move individuals and cause them to experience positive emotions [43], most kinds of music may be effective when used to regulate emotions positively. There may be individual differences in such effects [44], it is necessary to explore individual differences in the emotional adaptive function of music listening in further studies.

Vuoskoski, J., Eerola, T. The pleasure evoked by sad music is mediated by feelings of being moved. Front. Psychol. 2017, 8, 439. https://doi.org/10.3389/fpsyg.2017.00439

Eerola, T., Vuoskoski, J.K., Kautiainen, H., Peltola, H.R., Putkinen, V., Schäfer, K. Being moved by listening to unfamiliar sad music induces reward-related hormonal changes in empathic listeners. Ann. NY Acad. Sci. 2021, 1502, 121-131. https://doi.org/10.1111/nyas.14660

Vuoskoski, J., Eerola, T. Measuring music-induced emotion. A comparison of emotion models, personality biases, and intensity of experiences. Musicae Sci. 2011, 15, 159–173. https://doi.org/10.1177/1029864911403367

Point 7. Page 4, line 135: why this choice for cognitive judgments only? Please provide at least some motivation for this limitation.

Response 7: The part of your comment is Diener's conceptualization of subjective well-being, emphasizing that life satisfaction is cognitive rather than affective. Diener et al.'s Satisfaction with Life Scale, which is widely used around the world, measures an individual's cognitive judgments of own life. We would really appreciate your understanding.

Point 8. Page 6, line 221: why using bootstrapping technique here, given the large number of participants? PIease motivate a little.

Response 8: Thank you for your comment. Although we had a sufficient sample, we felt it would be best to use a bootstrapping method for this study. The advantages of the bootstrapping method are added as follows. (Line 174-176)

Bootstrapping, a non-parametric resampling method, can examine the mediating effect even if the sample distribution does not meet normality [35].

Point 9. Page 7, lines 230 ff: the relation between the text of this paragraph and table 3 is not immediately clear.

Response 9: Thank you for your advice. As you advised, we changed paragraph as below. (Line 247-260)

Additionally, the sequential double mediating effect of perceived stress and life satisfaction on the relation of hardiness and expectations for future life (H → PS → LS → EFL) was 0.041 (0.0240–0.0632) and significant. The sequential double mediating effect of music listening for negative emotion regulation and life satisfaction on the relation of hardiness and expectations for future life (H → ML → LS → EFL) was 0.010 (0.0005–0.0236) and significant.

We also examined whether there were any differences in the indirect effect sizes revealed in this study (Table 3). The effect sizes of the indirect paths of H → LS → EFL was greater than that of the indirect path of H → PS → LS → EFL and H → ML → LS → EFL. In addition, the double mediating effect size of perceive stress and life satisfaction on the relation of hardiness and expectations for future life (H → PS → LS → EFL) was significantly greater than that of music listening for negative emotion regulation and life satisfaction (H → ML → LS → EFL: −0.0535 ~ −0.0105).

Point 10. Page 7, table 3: please explain much better how to interpret the table. What does BC stand for? What is meant with Ind 1, Ind2. Not all readers are familiar with bootstrapping. Some more intuitive description of the why and how could help them to interpret the table. Could it make sense also not to use the letters A, B, C and to replace them with the initials of the terms (H = hardness; ML = music listening, PS = perceived stress; LS = life satisfaction, EFL = expectation for future life)? This is only a suggestion. The table should be also more readable by adding the significance levels (p-value or *, **, ***).

Response 10: Thank you for your advice. As you advised, we changed table as below. However, PROCESS Macro only presents significance using bootstrapping when analyzing indirect effect and the differences between indirect effects, and does not provide us p values. Please understand that we could not able to present p values and *** marks in the table. (Line 244-246)

Table 3. Indirect effects of the mediation model

Path

Effect

S.E

BC 95% CI

Total indirect effect

 0.194

0.026

 0.1427 ~ 0.2463

Ind1: H → ML → EFL

−0.035

0.085

−0.0207 ~ 0.0131

Ind2: H → PS → EFL

 0.005

0.011

−0.0162 ~ 0.0257

Ind3: H → LS → EFL

 0.141

0.022

 0.1002 ~ 0.1848

Ind4: H → ML → LS → EFL

 0.010

0.006

 0.0005 ~ 0.0236

Ind5: H → PS → LS → EFL

 0.041

0.010

 0.0240 ~ 0.0632

Ind1 – Ind2

−0.009

0.013

−0.0340 ~ 0.0172

Ind1 – Ind3

−0.144

0.024

 −0.1912 ~ −0.0985

Ind1 – Ind4

−0.014

0.010

−0.0358 ~ 0.0036

Ind1 – Ind5

−0.045

0.013

 −0.0718 ~ −0.0189

Ind2 – Ind3

−0.135

0.026

 −0.1872 ~ −0.0867

Ind2 – Ind4

−0.005

0.125

 −0.0310 ~ −0.0187

Ind2 – Ind5

−0.036

0.016

 −0.0707 ~ −0.0088

Ind3 – Ind4

 0.130

0.023

 0.0873 ~ 0.1756

Ind3 – Ind5

 0.099

0.023

 0.0555 ~ 0.1449

Ind4 – Ind5

−0.031

0.011

 −0.0535 ~ −0.0105

H = Hardiness, ML = Music listening for negative emotion regulation, PS = Perceived stress, LS = Life satisfaction, EFL = Expectations for future life, Ind: Indirect effect
